# DG-TTA: Out-of-Domain Medical Image Segmentation Through Augmentation, Descriptor-Driven Domain Generalization, and Test-Time Adaptation

**DOI:** 10.3390/s25175603

**Published:** 2025-09-08

**Authors:** Christian Weihsbach, Christian N. Kruse, Alexander Bigalke, Mattias P. Heinrich

**Affiliations:** 1Institute of Medical Informatics, University of Lübeck, 23562 Lübeck, Germany; mattias.heinrich@uni-luebeck.de; 2EchoScout GmbH, 23562 Lübeck, Germany; 3Dampsoft GmbH, 24351 Damp, Germany; 4Drägerwerk AG & Co. KGaA, 23558 Lübeck, Germany

**Keywords:** domain generalization, domain-invariant descriptors, test-time adaptation

## Abstract

**Highlights:**

**What are the main findings?**
Combining domain-generalized pre-training with test-time adaptation substantially enhances segmentation accuracy for medical images in out-of-domain scenarios.Generalizing image descriptors and intensity augmentation during training and adaptation outperformed complex existing methods in several scenarios.

**What is the implication of the main finding?**
We provide a powerful tool to achieve high segmentation accuracy out-of-domain, even without access to source data.Our efficient method is applicable to many top public 2D and 3D segmentation models trained on large, private datasets.

**Abstract:**

Applying pre-trained medical deep learning segmentation models to out-of-domain images often yields predictions of insufficient quality. In this study, we propose using a robust generalizing descriptor, along with augmentation, to enable domain-generalized pre-training and test-time adaptation, thereby achieving high-quality segmentation in unseen domains. In this study, five different publicly available datasets, including 3D CT and MRI images, are used to evaluate segmentation performance in out-of-domain scenarios. The settings include abdominal, spine, and cardiac imaging. Domain-generalized pre-training on source data is used to obtain the best initial performance in the target domain. We introduce a combination of the generalizing SSC descriptor and GIN intensity augmentation for optimal generalization. Segmentation results are subsequently optimized at test time, where we propose adapting the pre-trained models for every unseen scan using a consistency scheme with the augmentation–descriptor combination. The proposed generalized pre-training and subsequent test-time adaptation improve model performance significantly in CT to MRI cross-domain prediction for abdominal (+46.2 and +28.2 Dice), spine (+72.9), and cardiac (+14.2 and +55.7 Dice) scenarios (*p* < 0.001). Our method enables the optimal, independent use of source and target data, successfully bridging domain gaps with a compact and efficient methodology.

## 1. Introduction

Medical image analysis, particularly image segmentation, has made a significant leap forward in recent years with the advent of deep learning. However, changes in data distribution introduced by different input modalities or devices can lead to errors in the performance of deep learning models [1]. Since multiple imaging techniques are often required for disease identification, and treatment planning, and MRI devices especially offer broad flexibility in adjusting acquisition parameters, access to all of these different domains is usually infeasible. Consequently, trained models may produce inaccurate results when encountering unseen, out-of-domain data at test time [2].

Supervised fine-tuning and transfer learning can be used as a workaround to adjust networks to the unseen domain, as seen in [3,4]. Still, this would in turn require curating and labeling data again, which is often costly and time-consuming. Frequently studied approaches to overcome this effort utilize domain translation and unsupervised domain adaptation methods, but they require simultaneous access to both source and target data [5,6]. Accessing the source and target data jointly introduces a challenge, as the source data may be unavailable during model adaptation to the target domain. Source-free domain adaptation circumvents those restrictions and requires only target data during source-model adaptation. Here, some methods perform retraining on a larger set of target images to adapt models [7,8]. In practice, a single out-of-domain data sample is often given for which we want to obtain optimal results immediately. We target this setting in our study, facing the most challenging data constraints. In this setting, domain generalization techniques can be used to optimize the source model performance for `any’ unseen out-of-distribution sample. In computer vision, techniques such as domain randomization, multi-task training, random convolutions, masking autoencoders, and image mask obfuscation have been proven to enhance model generalization successfully [9,10,11,12,13]. Specific approaches have also been designed in the medical domain to build foundation models for CT and MRI processing, optimally augmenting training input for CT and MRI cross-domain segmentation, discriminating input domains for adaptive CT lesion segmentation, synthesizing CT and MRI inputs for robust training of segmentation models, and combining text report and image data for multi-organ CT segmentation [14,15,16,17,18]. Domain generalization is an ultimate goal to achieve, but to date, no universal solution that works robustly has been found. Test-time adaptation (TTA), as a complementary approach, optimizes the source model performance only for one or a limited number of samples [19,20,21,22,23,24,25,26].

Linking both approaches enables optimal separate use of source and target data, where domain generalization maximizes the base performance, and TTA can further optimize the result. Numerous methods to bridge domain gaps have already been developed, but they often require complex strategies and assumptions such as intertwined adaptation layers [24,25], indirect supervision tasks [22,27], prior knowledge about label distributions [21], assumptions on the distinctiveness of domains [6], or many consecutive steps [28].

### Contributions

We propose employing DG-TTA, a minimally invasive and compact approach that utilizes a robust augmentation–descriptor scheme during domain-generalized pre-training and TTA for high-performance medical image segmentation in unseen domains with significant domain gaps.
We propose combining DG pre-training and TTA to achieve optimal performance with minimum data requirements (DG-TTA).We introduce the use of the SSC feature descriptor, previously applied in image registration tasks for DG pre-training and TTA, and we demonstrate its superiority on small-scale datasets.We perform TTA with a lean self-supervision scheme to avoid auxiliary optimization tasks or the need for prior assumptions, as opposed to related TTA methods.The proposed methodological contributions are evaluated in several out-of-domain CT-to-MR segmentation scenarios.

The proposed methodological contributions were integrated into the state-of-the-art nnUNet segmentation framework [29] to obtain high-quality predictions of medical 3D images within a unified framework.

## 2. Materials and Methods

### 2.1. Study Design and Patients

We included data from five publicly available datasets in this retrospective study (see Figure 1 and Table A1). The contents of all datasets have been previously reported [30,31,32,33,34]. Those prior studies focused on data collection and the development of individual segmentation methods, whereas we aim to develop a universal method for segmentation in this study. Throughout the following paragraphs, we use abbreviated names for the BTCV, AMOS, TotalSegmentator (TS), MyoSegmenTUM spine (SPINE), and MMWHS datasets. Example images of the datasets are presented in Figure 2. From the mentioned datasets, cross-domain prediction settings are compiled, all targeting the challenging domain gap between CT source and MR target prediction (CT > MR). Data partitioning was performed randomly and kept throughout all evaluated methods for fair comparison.

### 2.2. Datasets

#### 2.2.1. BTCV: Multi-Atlas Labeling Beyond the Cranial Vault

The dataset [30] contains 30 labeled abdominal CT scans of a colorectal cancer chemotherapy trial with 14 organs: spleen (SPL), right kidney (RKN), left kidney (LKN), gallbladder (GAL), esophagus (ESO), liver (LIV), stomach (STO), aorta (AOR), inferior vena cava (IVC), portal vein and splenic vein (PSV), pancreas (PAN), right adrenal gland (RAG), and left adrenal gland (LAG). Data dimensions reach from 512 × 512 × 85 vox to 512 × 512 × 198 vox and fields of view from 280 × 280 × 280 ${\mathrm{mm}}^{3}$ to 500 × 500 × 650 ${\mathrm{mm}}^{3}$. We split the dataset into a 20/10 training/test set for our experiments, using a subset of ten classes that are uniformly labeled across all scans (Classes AOR, PSV, RAG, and LAG were omitted).

#### 2.2.2. AMOS: A Large-Scale Abdominal Multi-Organ Benchmark
for Versatile Medical Image Segmentation

The AMOS dataset [31] comprises CT and MRI scans from eight scanners with a similar field of view to the BTCV dataset, which includes patients with structural abnormalities in the abdominal region (e.g., tumors). Unlike the BTCV dataset’s organs, AMOS has additional segmentation labels for the duodenum, bladder, and prostate/uterus, but not for the PSV class.

#### 2.2.3. MMWHS: Multi-Modality Whole Heart Segmentation

This dataset [32] contains CT and MR images of seven cardiac structures: the left ventricle, right ventricle, left atrium, right atrium, the myocardium of the left ventricle, the ascending aorta, and the pulmonary artery. The CT data resolution is 0.78 × 0.78 × 0.78 ${\mathrm{mm}}^{3}$/vox. The cardiac MRI data were obtained from two sites using a 1.5 T scanner and reconstructed to achieve resolutions ranging from 0.80 × 0.80 × 1.00 ${\mathrm{mm}}^{3}$/vox to 1.00 × 1.00 × 1.60 ${\mathrm{mm}}^{3}$/vox.

#### 2.2.4. SPINE: MyoSegmenTUM Spine

This MRI dataset [33] contains water, fat, and proton density-weighted lumbar spine scans with manually labeled vertebral bodies (L1–L5). The field of view is 220 × 220 × 80 ${\mathrm{mm}}^{3}$, with a resolution of 1.8 × 1.8 × 4.0 ${\mathrm{mm}}^{3}$/vox.

#### 2.2.5. TS: TotalSegmentator, 104 Labels

The large-scale TS dataset contains CT images of 1204 subjects with 104 annotated classes. The annotations were created semi-automatically, with a clinician reviewing each one. The data was acquired across 8 sites on 16 scanners with varying slice thickness and resolution [34]. Unlike the SPINE dataset, this dataset includes the vertebral bodies and spinous processes in its class labels. Model predictions were corrected accordingly in a postprocessing step to obtain reasonable results for evaluation (see the following paragraph).

#### 2.2.6. Pre-/Postprocessing

We resampled all datasets to a uniform voxel size of 1.50 × 1.50 × 1.50 ${\mathrm{mm}}^{3}$/vox. For the SPINE task, we cropped the TS ground truth to exclude the spinous processes, using a mask dilated by five voxels around the proposed prediction in the TS > SPINE out-of-domain prediction setting to provide comparable annotations. Visualizations of the resulting ground truth labels can be seen in Section 3.

### 2.3. Related Work

#### 2.3.1. Domain Generalization

One way to improve model generalization is to increase the data manifold via augmentation [35]. Augmentations can comprise simple intensity-based modifications such as the application of random noise, partial corruption of image areas [12,13], randomly initialized weights [11,15], or differentiable augmentation schemes [16]. Generalization by domain randomization [9] leverages a complete virtual simulation of input data to provide broadly varying data [17]. Using specialized self-supervised training routines has also proven to be effective in improving model generalization [10,14].

#### 2.3.2. Test-Time Adaptation

Test-time adaptation (TTA) is performed in the target data domain and can be limited to a single target sample without access to the source data. The tent is an often-cited approach that adapts batch normalization layers of the network by minimizing prediction entropy [19]. Other works have successfully introduced auxiliary tasks [23] or priors to steer the adaptation, such as AdaMI [21]. RSA utilizes edge-guided diffusion models to translate images from the source to the target domain and selects the best-synthesized edge-image candidate based on the consistency of predictions [28]. Autoencoders capturing feature statistics can reduce implausible target segmentation outputs, such as TTA-RMI [23] or [24,25]. Approaches nearest to our proposed method utilize consistency-self-supervision schemes in combination with sample augmentation, but they introduce further model complexity through the addition of Mean Teacher or domain adversarial methods [6,36]. Many of the mentioned methods employ 2D models for image segmentation due to the memory requirements of the pipeline elements.

### 2.4. Proposed Method

We aim to leverage compact and effective domain-generalizing augmentation, as well as self-supervision during test-time adaptation, to enhance the performance of 3D segmentation models across domains. As shown in Figure 3, our method consists of two steps. The domain-generalized pre-training of the segmentation network involves using domain-generalizing techniques on the source image input (see the following paragraph). Later, our TTA strategy is employed on individual target domain samples and does not require access to the source data. Both steps are integrated into the state-of-the-art nnUNet segmentation framework [29].

### 2.5. Domain-Generalized Pre-Training on Source Data

Pre-training is performed on the labeled source training dataset $D_{train}={\left\{x_{s},y_{s}\right\}}_{s=1}^{l}$, l∈N, where $x_{s}$ and $y_{s}$ can also be patches. Recently, global intensity non-linear augmentation GIN [15] was introduced to improve model generalization. In GIN, a shallow convolutional network $g_{\rho }$ is re-initialized at each iteration by random parameters ρ and used to augment the input x. The augmented image is then blended with the original image, weighted by α:(1)$$\mathrm{GIN}\left(x\right)=\alpha \;g_{\rho }\left(x\right)+\left(1-\alpha \right)\;x$$

We propose combining GIN augmentation with self-similarity context (SSC) descriptors [37]. The approaches can be considered orthogonal, where GIN augmentation increases the input data manifold, and SSC features were designed to yield a robust generalized description. Our intuition is that GIN-augmented features effectively enrich the SSC descriptor space, thereby providing the network with meaningful input to generalize more effectively. Further explanations regarding this intuition can be found in the Appendix B in Figure A1. The SSC descriptor is defined as follows, with x (The CT and MR images used in our experiments do not necessarily follow a Gaussian distribution. The descriptor can generalize to images without prior assumptions about normality, as presented in Section 3) denoting the input image, p denoting the position of a central image patch, d denoting the position of neighboring patches within the neighborhood N, SSD denoting the sum of squared distances, and $\sigma_{N}^{2}$ denoting a noise estimate defined as the mean of all patch distances.(2)$$\mathrm{SSC}\left(x,p,d\right)=\exp\left(-\frac{SSD(x,p,d)}{\sigma_{N}^{2}}\right),\;p,d\in N,$$

The generalizing SSC descriptor aggregates distance measures of the neighborhood around an image patch, neglecting the image patch itself. The 6-neighborhood pattern N diagonally connects adjacent patches around the center patch, resulting in a mapping of $R^{1\times |\Omega |}\to R^{12\times |\Omega |}$ for voxel space Ω [37]. For our experiments, we use a patch size and patch distance of 1 vox.

### 2.6. Target Domain TTA

Our test-time adaptation method can now be applied to pre-trained models. For any given pre-trained model $f_{\theta }$ on the training dataset $D_{train}$, we aim to optimally adjust the weights to a single unseen sample from the target test set $D_{test}=\left\{x_{t}\right\}$ during TTA. Instead of adding complex architectures, we propose to use two augmentation functions, *A* and *B*, to obtain differently augmented images. In our case, *A* and *B* are solely comprised of a spatial augmentation function *S*; therefore, $A=S_{A}$ and $B=S_{B}$. The core idea of the method is to optimize the network to produce consistent predictions given two differently augmented inputs, where $x_{A,t}$ and $x_{B,t}$ are the spatially augmented images from function *A* and *B* (branch *A* and *B*), respectively:(3)$$\begin{array}{cccc} x_{A,t} & =A(x_{t}) & =S_{A}\left(x_{t}\right) & \;S_{A}:R^{\left|\Omega \right|}\to R^{\left|\Omega \right|} \end{array}$$(4)$$\begin{array}{cccc} x_{B,t} & =B(x_{t}) & =S_{B}\left(x_{t}\right) & \;S_{B}:R^{\left|\Omega \right|}\to R^{\left|\Omega \right|} \end{array}$$

Both augmented images are passed through the pre-trained network $f_{\theta }$:(5)$$\begin{array}{c} {\hat{y}}_{A,t}=f_{\theta }\left(x_{A,t}\right) \end{array}$$(6)$$\begin{array}{c} {\hat{y}}_{B,t}=f_{\theta }\left(x_{B,t}\right) \end{array}$$

Before calculating the consistency loss, both network predictions, ${\hat{y}}_{A,t}$ and ${\hat{y}}_{B,t}$, need to be mapped back to the initial spatial orientation for voxel-wise compatibility by applying the inverse transformation operation $S_{A}^{-1}$ and $S_{B}^{-1}$. In addition, a consistency masking $m_{c}\left(\;\cdot \;\right)$ is applied to filter inversion artifacts, with ζ indicating voxels that were introduced at the image borders during the inverse spatial transformation but are unrelated to the original image content. Inversion and masking yield the branch inversion functions $A^{-1}$ and $B^{-1}$:(7)$$\begin{array}{cc} m_{c}\left({\hat{y}}_{A,t},{\hat{y}}_{B,t}\right) & =\left[{\hat{y}}_{A,t}\neq \zeta \right]\wedge \left[{\hat{y}}_{B,t}\neq \zeta \right] \end{array}$$(8)$$\begin{array}{cc} A^{-1} & =m_{c}\circ S_{A}^{-1} \end{array}$$(9)$$\begin{array}{cc} B^{-1} & =m_{c}\circ S_{B}^{-1} \end{array}$$

We steer the network to produce consistent outputs by comparing them with the test-time loss $L_{TTA}$, defined as follows:(10)$$L_{TTA}=\ell \left({\hat{y}}_{A,t},{\hat{y}}_{B,t}\right)=\ell \left(A^{-1}\circ f_{\theta }\left(x_{A,t}\right),B^{-1}\circ f_{\theta }\left(x_{B,t}\right)\right)$$

As the loss function *ℓ*, we choose a Dice loss with predictions ${\hat{y}}_{A,t}$ and ${\hat{y}}_{B,t}$ given as class probabilities for all voxels in Ω, where *e* is a small constant ensuring numerical stability. |B| and |C| indicate the batch and channel sizes, respectively. ${\hat{y}}_{A,t,\omega }$ and ${\hat{y}}_{B,t,\omega }$ represent individual voxel values:(11)$$\ell \left({\hat{y}}_{A,t},{\hat{y}}_{B,t}\right)=1-\frac{1}{\left|B||C\right|}\overset{B,C}{\sum }\frac{\sum_{\omega }^{\Omega }2\;\cdot \;{\hat{y}}_{A,t,\omega }\;\cdot \;{\hat{y}}_{B,t,\omega }+e}{\sum_{\omega }^{\Omega }{\hat{y}}_{A,t,\omega }^{d}+{\hat{y}}_{B,t,\omega }^{d}+e},\;\omega \in \Omega$$

Selecting exponent d=2 ensures consistency in the Dice loss landscape, rather than d=1, which forces the network to additionally maximize the confidence of the prediction (see Figure 4). For spatial augmentation, we use affine image distortions on image/patch coordinates, which we found to be sufficient during our experiments.

#### Optimization Strategy

During TTA, only the classes *C* of interest are optimized for consistency (The classes of interest are an arbitrary choice, depending on the use case of the adaptation). To increase the robustness of predicted labels, we use an ensemble of three TTA models in the final inference routine of the nnUNet framework [29]. The AdamW optimizer was used with a learning rate of $\eta =1\;\times \;10^{-5}$, weight decay β=0.01, and no scheduling. We empirically selected a count of $N_{s}=12$ optimization steps throughout all of our experiments. Special caution has to be taken when applying test-time adaptation to models that require patch-based input. Since patch-based inference limits the field of view, the optimizer will adapt the model weights and overfit for consistency of the specific image region. Therefore, we accumulate gradients of $N_{p}=16$ randomly drawn patches during one optimization step.

### 2.7. Statistical Methods

In the following sections, segmentation quality is evaluated using the Dice score overlap metric (Dice) and the 95th percentile of the Hausdorff distance (HD95). The significance of TTA improvements is determined with the one-sided Wilcoxon Signed Rank test [38], with significance levels denoted as * *p* < 0.05, ** *p* < 0.01, and *** *p* < 0.001 (software used: Python 3.11, scipy 1.14.1).

## 3. Results

### 3.1. Experiment I: Abdominal CT/MR Cross-Domain Segmentation

In this experiment, we evaluate the performance of multiple base models and adapted models in an abdominal segmentation scenario. All base models were trained on source CT data (denoted by BS in the figures and tables). NNUNET denotes the standard model of the nnUNet pipeline [29] without specialized domain generalization capabilities. NNUNET BN denotes a nnUNet model with batch normalization layers. GIN, SSC, and GIN + SSC base models were pre-trained with the domain generalizing techniques described in Section 2.5. For comparison, we report the results of four related cross-domain methods mentioned in Section 2.3: Tent, TTA-RMI, RSA, and AdaMI [19,21,23,28]. Tent and AdaMI only required minor changes to the pipeline (loss and layers), and we have integrated them into the nnUNet pipeline. For the evaluation of TTA-RMI and RSA, we integrated the scenario data into the methods’ pipelines. For AdaMI, a class ratio prior needs to be provided, which we estimated by averaging the class voxel counts of the training dataset while we considered the same image field of view for the patch-based input. In addition to the base models’ performances, we report the adapted models’ performance after TTA (denoted by +A) and the significance of improvements compared to the NNUNET BS base model (reference). In the case of the batch normalization model NNUNET BN, we additionally evaluated adapting only the normalization layer parameters (+A-nor) or only the encoder (+A-enc). In all other experiments, all model parameters were adapted.

The results can be compared by referring to the boxplots in Figure 5 or Table 1, where the Dice similarity is presented. Table 2 presents Hausdorff distance results. For reference, we report the in-domain target data performance when training the NNUNET model on the target domain (no test data was included in target training).

The NNUNET base model achieves a mean Dice value of 32.0% when predicting across domains. This represents a drop of −52.6% compared to the NNUNET model when trained on the target data domain (reference model). Tent can outperform the reference model by +17.6% Dice, with significant improvements. TTA-RMI outperforms the reference model by +7.8% Dice. With RSA, we experience a decrease to 10.7% Dice. AdaMI performs best among the comparison methods, with a mean Dice value of 62.8% after adaptation.

Pre-trained models with generalization techniques achieve Dice scores of 76.3%, 76.1%, and 78.6% when predicting across domains (GIN, SSC, and GIN + SSC). Subsequent adaptation gains +1.1%, +1.6%, and −0.4% Dice. The highest mean performance is reached by the GIN + SSC pre-trained model (78.6%), whereas its adaptation results in a slight performance decrease (−0.4%). The HD95 distance can be reduced to a minimum of 17.2 mm after adapting the GIN + SSC model. Using our TTA scheme, the performance of non-generalizing pre-trained models improves significantly. The highest model internal improvement is reached for the NNUNET BN model when all model parameters are adapted during TTA (+64.0% Dice and −42.9 mm HD95). Adapting partial layers of the model results in lower gains. The Hausdorff distance measurements (HD95) in Table 2 reflect the changes in the Dice score in most cases. For Tent, an increase in the mean HD95 distance was measured, whereas Dice performance increased. This discrepancy can be explained by falsely predicted pixels that appeared in the outside regions of the image, far away from the organs’ centers.

Although substantial gains can be achieved with our method, there remains a gap of −5.9% Dice to the model that has been trained on the target domain data directly (target training, in-domain Dice score of 84.5% vs. our best-performing cross-domain model GIN + SSC 78.6%).

#### Runtime

Pre-training of the GIN-SSC model took 15.3 h and was comparable to the other model variants. Test-time adaptation of a single scan took approximately 1.7 min per fold on an Nvidia Tesla V100-SXM3-32GB GPU. For a three-fold ensemble, this accumulates to 5.2 min per scan.

### 3.2. Experiment II: Multi-Scenario CT/MR Cross-Domain Segmentation with DG-TTA

Building upon Experiment I, we demonstrate the efficacy of our method, leveraging the TS dataset’s 600 training samples as a strong basis in three segmentation tasks (all CT > MR): abdominal organ, lumbar spine, and whole-heart segmentation (TS > SPINE, TS > AMOS, TS > MMWHS MR). Unlike to the large TS dataset, we present results for the whole-heart segmentation task using only as few as 12 CT samples in model pre-training (MMWHS CT > MR).

Abdominal prediction across domains using the TS > AMOS datasets resulted in 64.1%, 68.4%, 81.4%, 79.0%, and 79.6% Dice similarity after adapting the models (see Figure 6). All adapted and generalized pre-trained models significantly increased the base model’s performance. Compared to the more limited BTCV training dataset tested in Experiment I (GIN + SSC + A, 78.6% Dice), training on the TS dataset yielded better top results (GIN + A, 81.4% Dice). The best mean Dice score for lumbar spine segmentation was reached by the GIN + SSC-adapted model (73.7% Dice). In the cardiac segmentation scenario, rich and low-sample training datasets were compared. For cross-domain prediction with the rich-sample pre-trained TS model, a best mean Dice score of 82.6% was reached with GIN augmentation. For the low-sample training MMWHS dataset, GIN+SSC+A reached the highest mean Dice of 71.5%. Visual results of the mentioned scenarios are depicted in Figure 7.

Table 3 summarizes the mean Dice and HD95 scores of all scenarios for the GIN, SSC, and GIN + SSC methods, along with their ranked scores. We found GIN + SSC + A to perform best across all scenarios, reaching a score rank of 1.9, outperforming SSC + A and GIN + A (ranks 3.3 and 3.5).

### 3.3. Experiment III: TTA Ablation Experiments

In this ablation experiment, the best configuration of our TTA scheme is evaluated using combinations of the individual TTA pipeline blocks: intensity augmentation, spatial augmentation, and gradient backpropagation of the self-supervised consistency Dice loss in the two branches, A and B (see Figure 8). Disabling or enabling the blocks of our pipeline resulted in 56 combinations. We selected one conventionally pre-trained base model (NNUNET BS) and one generalized pre-trained base model (GIN BS) to find an optimal configuration of our method in the abdominal segmentation task BTCV > AMOS.

Varying blocks for the NNUNET model results in up to +30.1% gains in the Dice score. For the GIN model, the maximum gain in Dice score is +1.6%, as this model is already generalizing well. TTA benefits from both intensity and spatial augmentation; however, no clear trend can be observed to reject specific configurations entirely (except for applying no augmentation at all, which consistently yields outputs for branches). Combining the ranks of the Dice gains of both models Figure 8a,b), we found that using gradients and backpropagation only in branch *A*, without intensity augmentation, and affine spatial augmentation in branches *A* and *B*, is an optimal trade-off between conventionally pre-trained and domain-generalized pre-trained models (see Figure 8c).

## 4. Discussion

Our experiments demonstrated that the GIN+SSC augmentation–descriptor combination is highly effective, particularly when using limited data samples in the pre-training stage of our two-step approach. If the domain-generalized pre-training does not result in sufficient target domain performance, our test-time adaptation scheme recovers weakly performing networks. This adaptation is crucial, as it is impossible to know the target domain properties a priori. Significant gains were achieved in all scenarios with the GIN+SSC combination, especially in the cardiac MMWHS CT > MR scenario, where we report improvements from +21.8% Dice for GIN+SSC over using GIN-only augmentation. The presented results are consistent across five challenging CT > MR out-of-domain prediction scenarios spanning abdominal, cardiac, and spine segmentation with small- and large-scale datasets from 12 to 600 training samples. Our approach can address many performant public segmentation models that have been trained on large, private, and unshareable datasets without the need to control their training regime for domain generalization directly. This flexibility is, moreover, beneficial because model providers would usually opt for a less generalizable model if this led to higher in-domain performance. Here, our method reaches improvements of up to +64.0% Dice for non-domain-generalized pre-trained models (see Section 3.1, model NNUNET BN).

### 4.1. Pertinent Findings in This Study

The proposed GIN+SSC augmentation–descriptor scheme outperforms augmentation-only and descriptor-only configurations in our pre-training and TTA pipeline with a best score rank of 1.9. Similar to earlier works, we updated the batch normalization layers of the models with our TTA scheme; however, the mean performance is higher when using consistency loss compared to the entropy-based formulation of Tent. Additionally, adapting all parameters is preferred over adjusting only single layers or parts of the model using our consistency scheme. Many cross-domain methods require 2D models, as the surrounding pipelines are complex and impose limits on the base model’s memory size. Our compact scheme can be used with 3D models and does not require prior assumptions.

### 4.2. Differences with Regard to Existing Literature

The TTA-RMI, AdaMI, and RSA methods evaluated in this study are tailored to specific setups, datasets, and their properties. Adaptation with AdaMI was successful but requires a class-ratio prior. This assumption is tough to fulfill in patch-based frameworks, where it is unclear what organ is visible and how large it will appear in the image region. We could not achieve high scores with RSA, which generates convincing source-style CT images from MR inputs; however, the predicted 2D masks are scattered in 3D space since the method does not include 3D convolutional layers.

In our experience, there is a tendency towards designing complex methods. This complexity is reflected in longer training and adaptation runtimes, such as 126.6 h for the domain translation and segmentation training pipeline and 82.0 min of adaptation for the method presented in [28]. Our proposed method required 15.3 h of training and 5.2 min for adaptation. It is compact and readily integrated into the well-established nnUNet pipeline [29], as it solely involves input-feature modifications and self-supervised test-time adaptation.

### 4.3. Limitations of the Technical Method

GIN augmentation alone can achieve sufficiently high performance out-of-domain when considering anatomies for which large pre-training datasets are available. Here, our TTA scheme yields only moderate additional gains (abdominal: +1.1% and +0.2%, lumbar spine: −0.4%, and cardiac: +1.2% Dice score). We empirically selected the number of adaptation epochs and observed further score gains or, occasionally, score drops for samples when the test-time adaptation continued beyond the specified epoch. A precise measure of convergence to stop the adaptation would be needed, but it is currently out of reach (since TTA has no ground-truth target to evaluate). We maintained the chosen number of epochs throughout all performed experiments and scenarios and are thus confident that this choice is reasonable for new scenarios as well. Apart from intensity-level domain gaps, domain gaps in image orientation and resolution of the target domain scans impose further difficulties for cross-domain predictions. To mitigate these issues, the pre-training of our source models could be further optimized by using multi-resolution augmentation in this regard, or images of the unseen target data could be reoriented to a standardized orientation.

## 5. Conclusions

In this study, we investigated the application of domain-generalized pre-training and augmentation-based test-time adaptation for robust cross-domain medical image segmentation. We found that models pretrained with a combination of generalizing augmentation and a generalizing feature descriptor can successfully predict across CT and MRI domains (GIN augmentation + SSC descriptor). An augmentation-based test-time adaptation scheme subsequently enhances segmentation performance in cases where the domain-generalized pre-training does not suffice. Substantial gains could be achieved with our GIN + SSC method for abdominal (+46.2 and +28.2 Dice), spine (+72.9), and cardiac (+14.2 and +55.7 Dice) scenarios (*p* < 0.001). These gains demonstrate that pre-trained models for volumetric medical segmentation can be effectively reused on target scans using a compact methodology that can be easily integrated into existing pipelines.

Our method enables the reuse of models pretrained on small-scale and large-scale CT datasets on MR images, which can differ vastly in intensity distribution. While cross-domain segmentation performance can be significantly improved, a gap remains compared to models trained directly on the target domain data (e.g., for the abdominal cross-domain prediction setting, our best-performing model, GIN + SSC, achieves 78.6% Dice, which is lower than the in-domain model’s performance of 84.5% Dice).

Our method effectively combines generalizing augmentation, a generalizing descriptor, and augmentation-based test-time adaptation to enable cross-domain medical image segmentation. The method proved effective in several scenarios, including abdominal, cardiac, and spine segmentation.

## Figures and Tables

**Figure 1 sensors-25-05603-f001:**
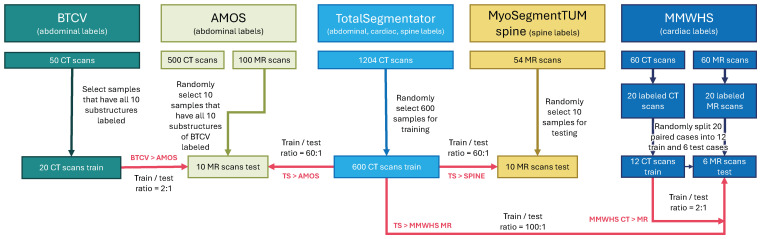
Study flowchart. Data from five publicly available datasets was included and combined to define several out-of-domain CT > MR prediction scenarios (the red arrows indicate their combinations) [30,31,32,33,34]. We randomly extracted subsamples for a source and target data ratio of at least 2:1. For the MMWHS dataset, we split the training and test data to include individual patients only (no paired data across training and testing).

**Figure 2 sensors-25-05603-f002:**
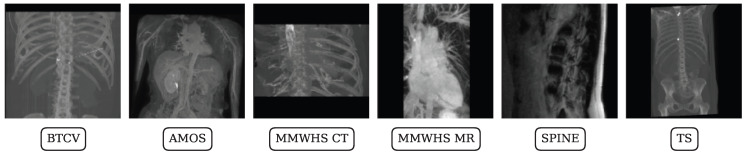
Sample images from the datasets used in this study, shown as maximum intensity projections.

**Figure 3 sensors-25-05603-f003:**
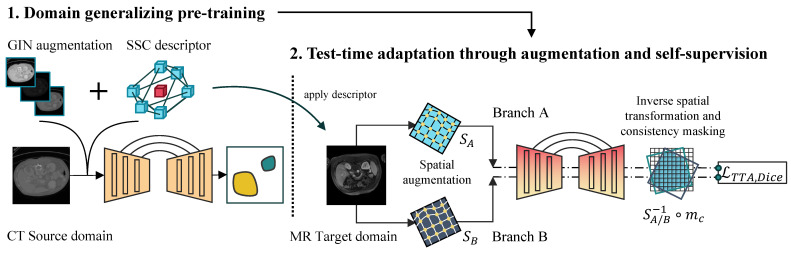
Our proposed method consists of two steps that, when combined, achieve optimal performance; however, they can also be used independently. Both steps rely on modifying input features to improve model generalization during training and enable unsupervised model adaptation at test time. (**Left**): Model pre-training with source domain data. We propose to use GIN augmentation [15] and the SSC descriptor [37] in this step. (**Right**): TTA is applied in the target data domain. Two different augmented versions of the same input are passed through the pre-trained segmentation network. The network weights are then optimized, supervised by the Dice loss, which guides the network to produce consistent predictions. After applying inverse spatial transformations, consistency masking is used to filter out non-matching regions.

**Figure 4 sensors-25-05603-f004:**
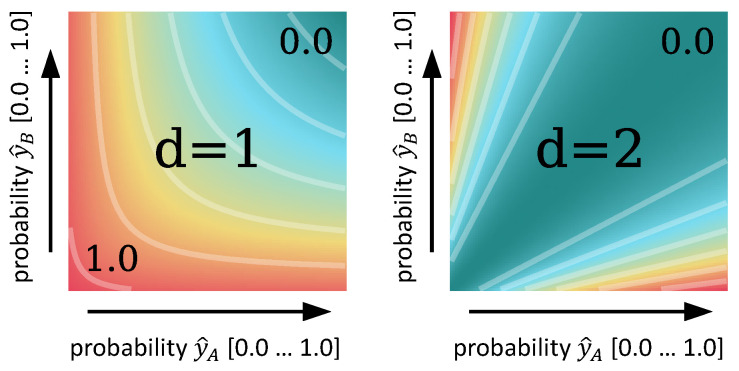
Dice loss landscapes given scalar probability values ${\hat{y}}_{A}$ and ${\hat{y}}_{B}$ for different exponents d=[1,2] in Equation (Equation 11). d=2 yields zero loss along the diagonal, which is favorable for consistency.

**Figure 5 sensors-25-05603-f005:**
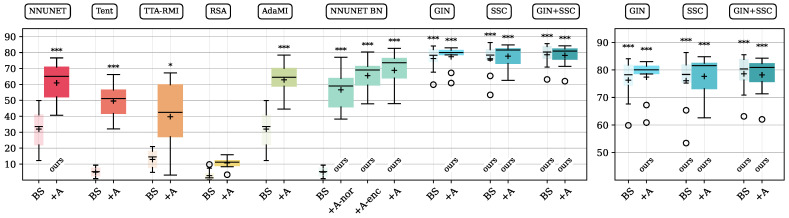
Base (BS) and adapted model (+A) performance of several methods bridging a CT > MR domain gap in abdominal organ segmentation. Ordinate shows Dice scores in percentage. Median (—) and mean (+) are indicated for boxes. The significance of improvement over the source NNUNET BS base model is shown above boxes (* *p* < 0.05; *** *p* < 0.001). The right part of the figure shows a zoomed-in view of the three rightmost proposed methods.

**Figure 6 sensors-25-05603-f006:**
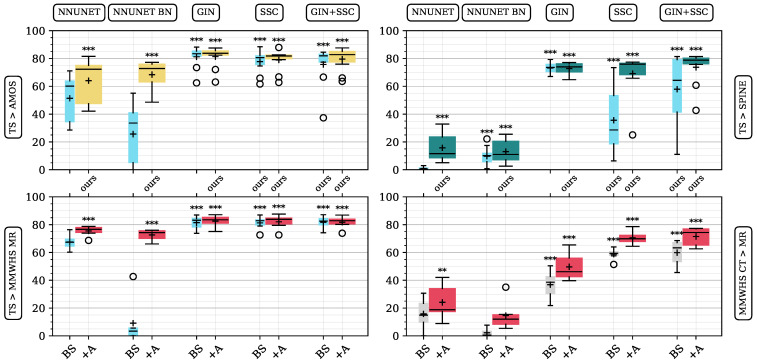
Base (BS) and adapted (+A) model performance given in Dice similarity percentage for several cross-domain prediction scenarios. Top row and bottom left: TS pre-trained models. Bottom right: MMWHS CT pre-trained models with only 12 training samples. Ordinate shows Dice scores in percentage. Median (—) and mean (+) are indicated for boxes. The significance of improvement over the source NNUNET BS base model is shown above boxes (** *p* < 0.01; *** *p* < 0.001).

**Figure 7 sensors-25-05603-f007:**
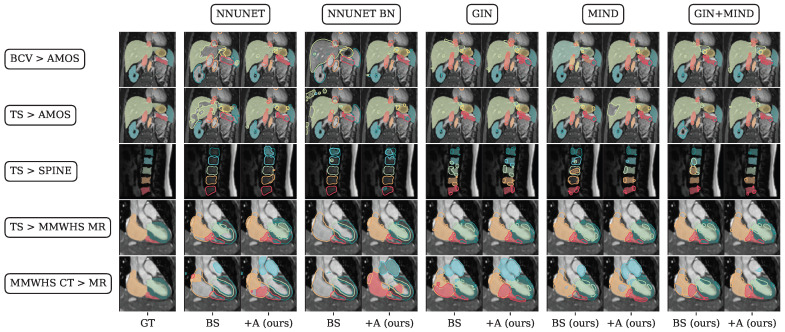
Visual results corresponding to the statistics in Figure 5 and Figure 6. The rows indicate the source and target datasets used, while the columns show the base (BS) or adapted (+A) models’ predictions. Ground truth is given in the leftmost column. Positively predicted voxels are shown in colors. Erroneous areas of predictions are marked with contours. The class colors for the abdominal task labels comprise spleen 
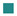
, right kidney 
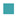
, left kidney 
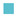
, gallbladder 
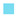
, esophagus 
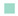
, liver 
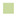
, stomach 
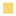
, aorta 
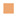
, inferior vena cava 
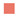
, and panreas 
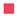
. Whole-heart class labels comprise the right ventricle 
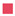
, right atrium 
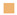
, left ventricle 
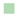
, left atrium 
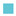
, and myocardium 
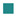
. Best viewed digitally.

**Figure 8 sensors-25-05603-f008:**
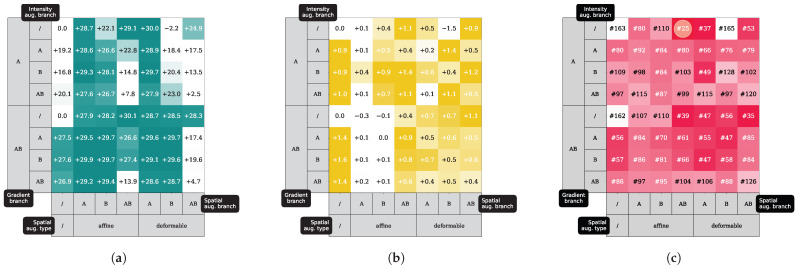
(**a**) Gains in Dice score on a non-domain-generalized pre-trained NNUNET model. (**b**) Gains in Dice score on a domain-generalized pre-trained GIN model. (**c**) Combined rank for configurations of (**a**,**b**). A, B and AB indicate whether augmentation or gradient backpropagation were enabled in branch A, branch B, or both branches (AB). The best rank is highlighted.

**Table 1 sensors-25-05603-t001:** Base (BS) and adapted model (+A) performance given in Dice similarity percentage of several methods bridging a CT > MR domain gap in abdominal organ segmentation. In the case of the batch normalization model NNUNET BN, we evaluated adapting only the normalization layer parameters (+A-nor) or the encoder (+A-enc), in addition to evaluating the adaptation of all parameters. Higher Dice values indicate better performance. Mean column corresponds to values in Figure 5. Class names abbreviated: spleen (SPL), right/left kidney (RKN/LKN), gallbladder (GAL), esophagus (ESO), liver (LIV), stomach (STO), aorta (AOR), inferior vena cava (IVC), and pancreas (PAN). Performance gains refer to the NNUNET BS model. Bold formatting indicates the best score per column.

Method	Stage		SPL	RKN	LKN	GAL	ESO	LIV	STO	AOR	IVC	PAN	Dice μ±σ	Gain
NNUNET	BS	Reference	40.2	21.9	15.9	24.9	22.9	76.0	34.3	26.4	21.8	35.3	32.0 ± 16.3	
	+A	ours	76.0	70.0	74.4	42.5	42.0	79.8	52.0	65.2	46.7	60.8	60.9 ± 13.6	+28.9
Tent	BS		0.0	0.0	0.0	0.0	0.3	43.7	0.0	2.3	0.5	1.5	4.8 ± 13.0	−27.2
	+A		68.7	68.4	80.3	30.2	25.3	50.3	52.4	47.0	30.3	43.0	49.6 ± 17.5	+17.6
TTA-RMI	BS		3.2	10.8	23.3	3.3	11.6	38.9	15.2	3.1	8.7	11.2	12.9 ± 10.5	−19.1
	+A		65.8	48.0	55.7	9.3	25.1	66.7	37.0	36.0	25.6	28.5	39.8 ± 17.9	+7.8
RSA	BS		5.5	4.5	4.3	0.0	0.0	4.4	7.3	1.2	0.0	0.3	2.7 ± 2.6	−29.2
	+A		12.5	14.1	18.6	0.9	2.4	24.5	5.3	10.0	15.8	2.9	10.7 ± 7.4	−21.3
AdaMI	BS		40.2	21.9	15.9	24.9	22.9	76.0	34.3	26.4	21.8	35.3	32.0 ± 16.3	
	+A		75.0	77.7	83.1	37.1	37.9	83.8	56.5	68.1	48.2	61.1	62.8 ± 16.7	+30.9
NNUNET BN	BS		0.0	0.0	0.0	0.0	0.3	43.7	0.0	2.3	0.5	1.5	4.8 ± 13.0	−27.2
	+A-nor	ours	62.8	79.2	80.1	32.8	26.5	74.0	67.6	52.5	34.5	56.3	56.6 ± 18.7	+24.6
	+A-enc	ours	80.1	87.1	88.4	34.6	30.9	83.1	73.9	68.8	43.8	64.8	65.5 ± 20.5	+33.6
	+A	ours	81.6	87.2	89.2	40.1	35.9	84.7	73.4	75.1	54.1	66.8	68.8 ± 18.3	+36.8
GIN	BS		81.9	90.4	91.9	63.7	47.8	92.7	73.2	80.9	72.1	68.1	76.3 ± 13.5	+44.3
	+A	ours	81.6	90.5	92.1	**72.3**	48.9	93.3	74.7	79.2	70.9	70.6	77.4 ± 12.6	+45.4
SSC	BS	ours	**83.6**	92.9	93.0	60.5	39.1	93.1	74.6	81.1	69.5	**73.2**	76.1 ± 16.1	+44.1
	+A	ours	83.4	91.2	92.9	68.1	48.6	91.4	74.4	83.3	71.3	72.2	77.7 ± 13.0	+45.7
GIN+SSC	BS	ours	83.0	**93.3**	**93.3**	65.9	**50.3**	**94.1**	**76.5**	83.9	**73.7**	71.9	**78.6 ± 13.3**	+46.6
	+A	ours	82.2	92.7	92.7	68.4	47.1	93.4	74.5	**84.8**	73.5	72.5	78.2 ± 13.6	+46.2
(NNUNET)	(Target training)		86.5	94.6	95.0	70.9	59.9	97.3	81.4	91.2	85.4	83.2	84.5 ± 11.1	+52.6

**Table 2 sensors-25-05603-t002:** Base (BS) and adapted model (+A) performance is given in the 95th percentile of the Hausdorff distance in mm (HD95) for several methods bridging the CT > MR domain gap in abdominal organ segmentation. Smaller distances indicate better performance. Class names abbreviated: spleen (SPL), right/left kidney (RKN/LKN), gallbladder (GAL), esophagus (ESO), liver (LIV), stomach (STO), aorta (AOR), inferior vena cava (IVC), and pancreas (PAN). Distance reduction refers to the NNUNET BS model (the more negative, the better). Bold formatting indicates the best score per column.

Method	Stage		SPL	RKN	LKN	GAL	ESO	LIV	STO	AOR	IVC	PAN	HD95 μ±σ	Reduction
NNUNET	BS	Reference	96.2	60.2	105.0	66.0	68.8	151.3	181.1	127.7	115.2	72.6	104.4 ± 38.1	
	+A	ours	70.9	36.5	58.2	93.8	51.5	158.1	191.4	126.3	116.5	72.8	97.6 ± 47.3	−6.8
Tent	BS		—	90.1	102.7	32.7	141.5	46.0	93.5	182.5	123.0	76.3	98.7 ± 43.7	−5.7
	+A		186.7	215.1	181.7	130.9	180.3	239.3	228.3	171.4	162.8	182.9	187.9 ± 30.5	+83.5
TTA-RMI	BS		105.9	101.0	92.6	69.4	149.9	114.3	163.3	101.9	109.6	96.8	110.5 ± 26.0	+6.1
	+A		67.9	47.6	84.8	65.9	122.0	93.5	103.4	99.7	112.9	76.9	87.4 ± 22.0	−17.0
RSA	BS		116.8	88.9	60.2	77.7	129.7	206.9	115.7	146.1	154.1	101.3	119.7 ± 40.2	+15.3
	+A		80.2	119.9	102.8	97.7	116.1	83.9	100.2	43.4	36.6	86.6	86.7 ± 26.4	−17.7
AdaMI	BS		96.2	60.2	105.0	66.0	68.8	151.3	181.1	127.7	115.2	72.6	104.4 ± 38.1	
	+A		39.3	46.1	44.7	85.5	40.9	174.4	196.1	106.2	97.1	56.2	88.7 ± 53.7	−15.8
NNUNET BN	BS		—	90.1	102.7	32.7	141.5	46.0	93.5	182.5	123.0	76.3	98.7 ± 43.7	−5.7
	+A-nor	ours	50.9	26.5	37.7	52.4	71.2	180.7	87.9	110.5	50.3	35.2	70.3 ± 44.1	−34.1
	+A-enc	ours	27.4	11.8	9.6	43.5	68.5	157.5	56.4	57.7	37.3	17.4	48.7 ± 41.1	−55.7
	+A	ours	38.1	54.0	42.6	50.4	33.0	134.6	70.2	82.9	31.6	20.2	55.8 ± 31.7	−48.6
GIN	BS		**9.5**	4.8	4.5	10.5	43.2	59.8	30.1	52.9	35.8	28.0	27.9 ± 19.1	−76.5
	+A	ours	18.1	9.8	4.7	8.5	50.6	57.9	47.4	77.7	33.0	63.1	37.1 ± 24.6	−67.3
SSC	BS	ours	19.5	3.7	4.3	9.5	19.0	**20.5**	26.5	55.3	38.3	37.5	23.4 ± 15.6	−81.0
	+A	ours	33.9	54.8	4.1	7.9	41.8	43.7	41.4	48.5	41.4	22.3	34.0 ± 16.2	−70.4
GIN+SSC	BS	ours	24.5	**3.5**	**3.9**	8.5	**10.7**	21.4	**20.0**	55.7	19.6	11.5	17.9 ± 14.4	−86.5
	+A	ours	42.3	3.8	4.1	**7.5**	11.2	23.8	23.7	**27.5**	**17.4**	**10.9**	**17.2 ± 11.6**	−87.2
(NNUNET)	(Target training)		2.9	6.2	7.7	6.2	7.0	18.8	19.2	6.3	5.3	8.4	8.8 ± 5.3	−95.6

**Table 3 sensors-25-05603-t003:** Mean base (BS) and adapted (+A) model performance for GIN, SSC, and GIN + SSC methods summarized for all evaluated scenarios. Performance given in Dice in percentage and the 95th percentile of the Hausdorff distance (HD95) in mm. Rank of the scores per group is given in brackets. The combined score rank is given in the last column of the lower table group (mean of ranks across all scores per method). Bold formatting indicates the best score per column.

Method	Stage		TS > AMOS Dice	TS > AMOS HD_95_	TS > SPINE Dice	TS >SPINE HD_95_	TS > MMWHS MR Dice	TS > MMWHS MR HD_95_
NNUNET	BS	Reference	51.4	68.2	0.8	59.2	67.6	34.8
GIN	BS		81.2 (2)	**11.8 (1)**	73.2 (2)	11.7 (3)	81.4 (5)	11.1 (6)
	+A	ours	**81.4 (1)**	14.2 (3)	72.8 (3)	9.7 (2)	**82.6 (1)**	10.2 (5)
SSC	BS	ours	77.7 (5)	16.7 (5)	35.6 (6)	22.8 (6)	81.3 (6)	8.9 (2)
	+A	ours	79.0 (4)	22.6 (6)	69.1 (4)	13.3 (4)	82.0 (2)	**8.5 (1)**
GIN+SSC	BS	ours	75.7 (6)	14.8 (4)	57.9 (5)	15.3 (5)	81.8 (4)	10.1 (4)
	+A	ours	79.6 (3)	12.1 (2)	**73.7 (1)**	**9.4 (1)**	81.8 (4)	9.4 (3)
**Method**	**Stage**		**BTCV > AMOS** **Dice**	**BTCV > AMOS** **HD_95_**	**MMWHS CT > MR** **Dice**	**MMWHS CT > MR** **HD_95_**		**COMBINED** **SCORE RANK**
NNUNET	BS	Reference	32.0	104.4	15.8	147.8		
GIN	BS		76.3 (5)	27.9 (4)	36.8 (6)	85.6 (6)		4.0
	+A	ours	77.4 (4)	37.1 (6)	49.7 (5)	77.5 (5)		3.5
SSC	BS	ours	76.1 (6)	23.4 (3)	58.8 (4)	53.0 (4)		4.7
	+A	ours	77.7 (3)	34.0 (5)	70.5 (2)	25.4 (2)		3.3
GIN+SSC	BS	ours	**78.6 (1)**	17.9 (2)	59.9 (3)	47.9 (3)		3.7
	+A	ours	78.2 (2)	**17.2 (1)**	**71.5 (1)**	**17.8 (1)**		**1.9**

## Data Availability

The open-source code of the presented method is available at: https://github.com/multimodallearning/DG-TTA (accessed on 31 August 2025). The original data presented in the study are openly available: BTCV [dataset] [30] AMOS [dataset] [31] MMWHS [dataset] [32] MyoSegmenTUM spine [dataset] [33] TotalSegmentator [dataset] [39].

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
