# Peer review of "DG-TTA: Out-of-Domain Medical Image Segmentation Through Augmentation, Descriptor-Driven Domain Generalization, and Test-Time Adaptation"

_sensors, 2025, doi:10.3390/s25175603_

Round 1

Reviewer 1 Report

Comments and Suggestions for Authors
  1. Paper lacks a theoretical foundation for why the combination of GIN and SSC descriptors works well.
  2. Computation cost need to be discussed, if passible include the qualitative results.
  3. These research article is very close to your work, you need cite it: "Optimization of deep learning-based denoising for arterial spin labeling: Effects of averaging and training strategies". 

Author Response

We want to thank the reviewer for taking the time to review our manuscript. We tried to address the mentioned points as follows:

Comment 1: The Paper lacks a theoretical foundation for why the combination of GIN and SSC descriptors works well. 

Response 1: Thank you for bringing this to our attention. Our intuition for why GIN+SSC works well was described in section "2.5. Domain-generalized pre-training on source data". This description was relatively short and without further evidence. We have added a new figure (Fig. A2) to Appendix B, which displays the image intensity distribution of GIN and GIN+SSC augmented images. The figure shows that GIN and SSC work in a perpendicular manner (diversification of the intensity profile for GIN vs. extraction of domain invariance for SSC), but that the domain-invariant representation of the SSC descriptor can be diversified slightly when GIN augmentation is applied prior to extracting the SSC features.

Comment 2: Computation cost needs to be discussed, if possible include the qualitative results

Response 2: We fully understand your concern and have added computational resources and runtime metrics to section 3.1. Experiment I: Abdominal CT/MR cross-domain segmentation, paragraph "Runtime" (line 286).

The presented metrics were also included in the discussion under Section 4. Discussion, "Differences with regard to existing literature" and set in contrast with runtimes of competing methods (line 366).

Comment 3: This research article is very close to your work, you need to cite it: "Optimization of deep learning-based denoising for arterial spin labeling: Effects of averaging and training strategies"

Response 3: We appreciate your suggestion to include the mentioned reference in our manuscript and have carefully reviewed the suggested publication. Although the reference presents interesting findings, we are concerned that it does not quite align with the scope of our manuscript.

The suggested reference presents a deep learning-based denoising method for arterial spin labeling in MRI. While the methodology also employs a U-Net-based architecture comparable to our network architecture, the target task and involved domains in our manuscript and the suggested reference differ. Our target task involves image segmentation, whereas the suggested reference denoises images to produce images with a better SNR (primarily working in the image space). Our manuscript targets cross-domain scenarios from CT to MR, whereas the suggested reference is tailored to MRI.

The closest match of content between the suggested reference and our manuscript is in our opinion section, "2.5.3 Generalizability of DL denoising". Here, the authors of the suggested reference describe the data split procedure for training and testing data, which should be performed using deep learning methods to test for model generalization. We also ensured that training and test data were not mixed in our experiments to evaluate whether the model's performance extends beyond training.

We would like to emphasize that our methodology also generalizes to data across multiple domains, whereas the suggested reference combined multiple datasets acquired by different scanners into a single large dataset, which was then split for training and testing.

If we have misinterpreted the suggested reference, we are open to discussing this further (ASL imaging is not our area of expertise).

Reviewer 2 Report

Comments and Suggestions for Authors

The paper "DG-TTA: Out-of-domain Medical Image Segmentation through Augmentation and Descriptor-driven Domain Generalization and Test-Time Adaptation" addresses a relevant issue: performing segmentation using neural networks for different types of medical images. This is very important, as large pre-trained networks allow for segmentation of specific types of images, often obtained under similar conditions. Regarding the problem being solved, it can be said that the article is suitable in theme for the journal Sensors.

The preceding works are well described, and the problem of needing to perform segmentation for different data, including those obtained by various methods, is outlined. References to recent research further confirm the relevance of the work. I would recommend providing a more detailed description of the works cited in lines 7-16, specifying the types of data for which the researchers perform segmentation.

The study undoubtedly has novelty, which lies in the use of generalizing data augmentation combined with a generalizing feature descriptor for segmentation of medical images across different domains. It has been demonstrated that the combination of GIN augmentation and the SSC descriptor is highly effective, especially with a limited number of data samples during the pre-training phase.

The article thoroughly describes the methods used in the research and their justification. Datasets and neural network models on which the authors improved and developed their algorithms are provided.

The figures are well executed, clearly displaying the algorithm's concept and effectively visualizing the results and errors.

The results are presented well and in detail. The combination of GIN+SSC methods achieved improvements in all scenarios. In the CT > MR scenario for cardiac images from the MMWHS, the improvement was over +21.8 % Dice compared to using only GIN augmentation. Positive results are confirmed across five CT > MR out-of-domain scenarios, covering segmentation of abdominal organs, the heart, and the spine. Both small and large datasets were used for training.

Overall, the results are clearly described and well visualized. The approach can be applied to large private and unshareable datasets.

I would recommend this article for publication with the following minor remarks:

In the conclusions, I believe it is advisable to add how much the accuracy of segmentation differs with your approach compared to training networks for a specific dataset, as your approach allows for segmentation of images from different domains.

It may be helpful to include the computational resources you applied in the methods section.

A comma is missing in the sentence (line 74) before "whereas."

Author Response

We thank the reviewer for taking the time to review our manuscript. We tried to address the mentioned points as follows:

Comment 1: I would recommend providing a more detailed description of the works cited in lines 7-16, specifying the types of data for which the researchers perform segmentation.

Response 1: We would like to thank you for pointing out this issue. We understand that the explanation of the provided references was very short. Lines 7-16 of our submitted manuscript do not actually contain cited works (they contain parts of the highlights and abstract text). However, we identified the range of cited works [7-16] in line 54 and inferred that you intended to suggest further explanation for those references. We thus added explanations about the content of the works, as well as the type of data used in the referenced studies.

Comment 2: I believe it is advisable to add how much the accuracy of segmentation differs with your approach compared to training networks for a specific dataset, as your approach allows for segmentation of images from different domains.

Response 2: Thank you for your suggestion. In Experiment 1, we already presented an in-domain target training score at the bottom of the data tables. We acknowledge your point that it is essential to compare the cross-domain segmentation and in-domain segmentation performance. To emphasize the presented scores, we added a paragraph about cross-domain and in-domain performance in Section 3.1. Experiment I: Abdominal CT/MR cross-domain segmentation (line 281) and 5. Conclusion (line 401).

Comment 3: It may be helpful to include the computational resources you applied in the methods section.

Response 3: We fully understand your concern and have added computational resources and runtime metrics to section 3.1. Experiment I: Abdominal CT/MR cross-domain segmentation, paragraph “Runtime” (line 286).

The presented metrics were also included in the discussion under Section 4. Discussion, “Differences with regard to existing literature” and set in contrast with runtimes of competing methods (line 366).

Comment 4: A comma is missing in the sentence (line 74) before “whereas.”

Response 4: Thank you for the in-depth review of our manuscript. We rechecked the grammar and spelling of the entire manuscript to enhance readability.

Reviewer 3 Report

Comments and Suggestions for Authors

Include the following reference.

Luna Lozoya, R.S.; Ochoa Domínguez, H.d.J.; Sossa Azuela, J.H.; Cruz Sánchez, V.G.; Vergara Villegas, O.O.; Núñez Barragán, K. Cross-Domain Transfer Learning Architecture for Microcalcification Cluster Detection Using the MEXBreast Multiresolution Mammography Dataset. Mathematics 2025, 13, 2422. https://doi.org/10.3390/math13152422

In the introduction section please make clear the contributions of the paper.

This reviewer cannot see clearly the objective of the research, please make it clear in the introduction section.

The hypothesis is clear from lines 58 to 60.

In Subsection 2.2.1 include a figure with sample images (the volumetric images can be represented using the maximum intensity projection). Same observation for sections 2.2.2, to 2.2.5 and 2.2.6. In the last one please use the original image and the processed image in the Figure.

In line 123 to substantiate the comment, please provide the appropriate reference.

Equation 2: Please explain the meaning of SSD(·). This reviewer assumes it is a distance measure; if so, this should be made clear in the manuscript.

Equation 2: Please clarify that the images follow a Gaussian distribution, or explain how the approach can be generalized to other distributions, such as the Poisson distribution (PET events acquisition).

Please use standard notation, as expressions like SA/B can be confused with the augmented input B given the augmented input A. After reading lines 174–178 several times, this reviewer understood that it refers to separated images. It can be expressed more clearly as: “SA and SB are the augmented images A and B, respectively.”

Please ensure that all notations used in the equations are defined immediately before or after their first occurrence.

Conclusion section is too short to understand what authors did. This reviewer recommends to include the following points (maybe some of them are not relevant):

Briefly remind the reader of what you set out to do and why.
Keep it concise — one or two sentences is enough.
Highlight only the main results, not every detail.
Link results back to the research questions or hypotheses.
If appropriate, mention whether they support or challenge previous studies.
Explain why your findings matter.
Indicate their practical, theoretical, or methodological contributions.
Connect to real-world applications, if relevant.
Acknowledge constraints in scope, methodology, or data.
Keep it brief, without undermining your results.
End with a concise, impactful takeaway.
Avoid introducing new data or arguments here.

The conclusion should not be overly limited, yet it is recommended to extend to roughly 10–20 lines to mention the recommended points.

Many claims in the text lack proper references.

The reference section should be updated.

Author Response

We thank the reviewer for taking the time to review our manuscript. We tried to address the mentioned points as follows:

Comment 1: Include the following reference. Luna Lozoya, R.S.; Ochoa Domínguez, H.d.J.; Sossa Azuela, J.H.; Cruz Sánchez, V.G.; Vergara Villegas, O.O.; Núñez Barragán, K. Cross-Domain Transfer Learning Architecture for Microcalcification Cluster Detection Using the MEXBreast Multiresolution Mammography Dataset. Mathematics 2025, 13, 2422. https://doi.org/10.3390/math13152422

Response 1: We would like to thank the reviewer for bringing this interesting reference regarding transfer learning to our attention. We integrated it into the new version of our manuscript.

Comment 2: In the introduction section, please make clear the contributions of the paper. This reviewer cannot see clearly the objective of the research. Please make it clear in the introduction section.

Response 2: We agree with the suggestion and added a contribution subsection that highlights the contributions of our work (line 73).

Comment 3: In Subsection 2.2.1 include a figure with sample images (the volumetric images can be represented using the maximum intensity projection). Same observation for sections 2.2.2, to 2.2.5 and 2.2.6. In the last one please use the original image and the processed image in the Figure.

Response 3: We would like to thank the reviewer for suggesting the use of maximum-intensity projected sample images to better describe the datasets used in our study. We believe that this is an excellent idea. We added an example image of every dataset to a new figure in the manuscript (page 5, top).

Comment 4: In line 123, to substantiate the comment, please provide the appropriate reference.

Response 4: We agree. We added a reference supporting our comment that augmentation is beneficial for medical image segmentation (the U-Net paper [35] mentions network architecture and data augmentation for improved segmentation performance). 

Comment 5: Equation 2: Please explain the meaning of SSD(·). This reviewer assumes it is a distance measure; if so, this should be made clear in the manuscript.

Response 5: We are grateful that the reviewer took the time to review the entire manuscript, including the equations. We ensured that every symbol is now introduced directly before or after any equation on its first occurrence.

Comment 6: Equation 2: Please clarify that the images follow a Gaussian distribution, or explain how the approach can be generalized to other distributions, such as the Poisson distribution (PET events acquisition).

Response 6: This is an interesting remark. The presented method does not assume a specific intensity distribution, and we added a comment regarding that to the manuscript as a footnote. We kindly refer to the newly integrated Figure A2 in Appendix B, which presents the image intensity distributions of a CT and an MR image.

Comment 7: Please use standard notation, as expressions like SA/B can be confused with the augmented input B given the augmented input A. After reading lines 174–178 several times, this reviewer understood that it refers to separated images. It can be expressed more clearly as: "SA and SB are the augmented images A and B, respectively."

Response 7: We appreciate that the reviewer took the time to read the paragraphs several times to grasp the content thoroughly. We have addressed this point and removed any confusing A/B notation from the sentences and equations (lines 200-220).

Comment 8: Please ensure that all notations used in the equations are defined immediately before or after their first occurrence.

Response 8: We appreciate that the reviewer thoroughly reviewed the entire manuscript, including the equations. We ensured that every symbol is now introduced directly before or after its first occurrence in an equation.

Comment 9: Conclusion section is too short to understand what authors did. This reviewer recommends to include the following points (maybe some of them are not relevant):

Response 9: We appreciate the reviewer's suggestion, which also included a brilliant scheme to enhance our conclusion. We extended our conclusion, taking into account the scheme that was provided.

Comment 10: Many claims in the text lack proper references. The reference section should be updated.

Response 10: We agree. We recognized the above suggestions for missing references and updated the reference section accordingly.

Round 2

Reviewer 3 Report

Comments and Suggestions for Authors

The required corrections have been satisfactorily addressed.